

# Comment on "Glauert's optimum rotor disk revisited – a calculus of variations solution and exact integrals for thrust and bending moment coefficients" by Tyagi and Schmitz (2025)

by

J. Gordon Leishman, Ph.D., D.Sc.(Eng.), F.R.Ae.S.
Professor Emeritus, Dept. of Aerospace Engineering, University of Maryland.
Distinguished Professor, Dept. of Aerospace Engineering, Embry-Riddle Aeronautical University.

This paper (Tyagi & Schmitz, 2025a) revisits the momentum theory problem for a windmill (i.e., a wind turbine) with apparent mathematical rigor and pedagogical eloquence. While it ultimately offers a seemingly mathematically rigorous reformulation, it contributes limited new insight to the wind energy field. Recasting Glauert's "optimum rotor" formulation using the calculus of variations and L'Hôpital's rule may suggest an air of analytical sophistication. Yet, it is little more than a formal re-derivation that does not clarify the underlying physical principles for engineering use. The authors introduce no new physical interpretation, favoring abstract derivations over engineering context. Although mathematically elaborate, their derivation of "exact" integrals for thrust and bending moment coefficients has limited practical relevance and does not materially advance rotor theory or wind turbine engineering.

The authors first examine the condition where the tip speed ratio $\lambda \to \infty$, and then find the power coefficient $C_P$ and the bending moment coefficient $C_{\mathrm{Be}}$. Physically, this corresponds to a scenario where the rotor is spinning much faster than the incoming wind speed, such that tip speed effects dominate over the relative inflow. In this regime, the axial velocity becomes negligible, and the flow approaches a purely tangential direction at the blade elements. The authors incorporate swirl into their equations through the angular induction factor; however, in the high-$\lambda$ limit, they assume $a' \to 0$, implying that the turbine imparts no swirl and, consequently, no torque. This issue introduces a fundamental contradiction in their theory, i.e., a turbine cannot extract power without torque. While the integrals they have derived may be mathematically consistent, they do not apply to a physically realizable situation. The authors neither resolve this mathematical inconsistency nor acknowledge this physical reality, which critically undermines the credibility of their high-$\lambda$ results. However, and more importantly, modern wind turbines are not designed to operate at very high tip-speed ratios, and this regime is generally avoided in practice. Indeed, under these conditions, the turbine blades would imply twist and chord distributions that are completely impractical.

In the low tip speed ratio limit ($\lambda \to 0$), the authors next derive exact integrals for $C_P$, $C_T$, and $C_{Be}$ using repeated applications of L'Hôpital's rule to resolve singularities. While these derivations may be formally correct, the physical context is again fundamentally flawed. In this regime, the turbine is either stationary, operating with blade stall, or in a transient startup phase, conditions under which no meaningful power is extracted, and the aerodynamic loads are dominated by separated and unsteady flow. In effect, the turbine behaves more as a high-drag device than an energy-extracting turbine; it operates in the turbulent wake state, which means that it obstructs the flow rather than extracting energy from it, thereby violating key assumptions of the simple one-dimensional flow model they have used. Under these conditions, the flow field is highly three-dimensional, non-uniform, and often dominated by large blade section angles of attack, rendering the momentum theory assumptions of steady, axisymmetric, and uniform flow with a constant pressure jump entirely inapplicable. Moreover, from a practical standpoint, modern turbines are designed to avoid operation in this regime altogether, typically idling or feathering at low wind speeds. The analytical effort invested by the authors in characterizing this physically irrelevant limit offers no guidance for turbine design, control strategy, or performance optimization.

Using the calculus of variations to rederive Glauert's third-order polynomial equation offers a modest pedagogical novelty. The expressions for $C_T$ and $C_{Be}$ may be formally "new" in closed form. Still, they are derived under conditions so idealized as to be irrelevant to any form of practical wind turbine engineering. Furthermore, the paper fails even to acknowledge the real possibility of tip losses, finite blade count, profile drag, wake expansion, or non-uniform and



yawed inflow. These are factors of higher priority in wind engineering practice. Indeed, in Glauert's original theory, the limiting behavior has not generally been regarded as a theoretical or practical limitation in wind energy research. Their so-called "math problem," therefore, is only one of their invention.

Additional issues arise in the presentation of their results. Several equations (e.g., Eqs. 35, 48, and 50) include numerical constants such as 2.5457 and $-13.3272$ without explanation or derivation. While not strictly erroneous, this practice reduces transparency and reproducibility. Providing explanations or citing how these constants were computed would have improved the clarity and independent reproducibility of their results. Without clearly stated methods or symbolic groundwork, the derivations appear procedural rather than intellectually motivated, making it more difficult to assess and reproduce the results independently. Indeed, despite the apparent technical competence of the derivations, their work remains disconnected from the practical challenges and standards of modern wind turbine research. There is no comparison to empirical data, computational results, or reconciliation with other findings. Their model assumes a wind turbine with an infinite number of blades that have a continuously optimal span loading at any $\lambda$. This is a mathematical abstraction that has no place in any realistic wind turbine analysis, particularly in the high- and low-$\lambda$ regimes that the authors specifically emphasize.

While their mathematical work appears internally consistent within the constraints of an idealized model, the model's assumptions physically break down in precisely those regions where their work attempts to provide insight. The authors do not extend Glauert's theory in a way that adds engineering value or new physical understanding. While the work may be of limited academic interest to those studying the historical development of rotor theory, it certainly falls short of the novelty, applicability, and physical relevance expected of contributions to *Wind Energy Science*. Instead, by clinging to an idealized and largely irrelevant theoretical framework, the authors ultimately diverge from the practical context of Glauert's original contributions. Their work most certainly does not "unlock new possibilities in wind turbine design that Hermann Glauert did not consider" (Sliman, 2025). Therefore, their framing is not aligned with established physical understanding and practical engineering considerations.

Finally, it is important to note that the central derivations, coefficient expressions, and conclusions of this article closely match those previously published by the same authors in a publicly accessible conference paper (Tyagi & Schmitz, 2025b). That earlier published work contains essentially the same analytical development, including the polynomial forms, integration techniques, convergence analysis, and variational formulation. This 2025 *Wind Energy Science* article does not cite or acknowledge the prior conference version, which is a notable omission that warrants editorial attention. Of greater concern is that both papers contain substantial verbatim reuse of text, structure, and phrasing. The core mathematical derivations are essentially the same, and the conclusions are restated with only minor editorial variation. This unacknowledged repetition represents substantial overlap with a prior publication; clarification of the relationship between the two works and appropriate citation would have been helpful. While this journal article appears more refined on the surface, the failure to cite a substantially overlapping publication, particularly one with broad public accessibility, suggests a need for clarification regarding attribution and transparency per journal policy.

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
