# Peer review of "Comment on "Glauert's optimum rotor disk revisited – a calculus of variations solution and exact integrals for thrust and bending moment coefficients" by Tyagi and Schmitz (2025)"

_Wind Energy Science, 2025_

## Referee Comment (RC1)

Prof. Dr. A.P. Schaffarczyk                                    Kiel, Germany, 2025 Sept 04

Kiel University of Applied Sciences

Faculty of Mechanical Engineering

**Review of manuscript WES-2025-105**

Comment on "Glauert's optimum rotor disk revisited – a calculus of variations solution and exact integrals for thrust and bending moment coefficients" by Tyagi and Schmitz (2025)

by

J. Gordon Leishman, Ph.D., D.Sc.(Eng.), F.R.Ae.S.
Professor Emeritus, Dept. of Aerospace Engineering, University of Maryland.
Distinguished Professor, Dept. of Aerospace Engineering, Embry-Riddle Aeronautical University

1.) General Comments

The author comments on a peer-reviewed and accepted paper of Tyagi and S. Schmitz on Glauert's general momentum theory of wind turbine rotors. A thorough discussion of this model, including its (sometimes hidden) assumptions and limitation was published earlier in [1]. The author criticizes that "no new physical interpretation" has given, "favouring abstract derivations over engineering context". In a last paragraph he notes, that parts of the paper have been previously published which has not been mentioned in the paper under comment.

It must be noted that – to the knowledge of the referee – all commercial or scientific BEM codeds used for wind turbine analysis and/or design use Glauert's approach including use of a'. An exception may be found in [3].

2.) Specific Comments

a) Paragraph 1, Lines 4 to 11
Calculus of Variation has a long and important history in theoretical, engineering and fluid mechanics. It can only be speculated why Glauert did not use this elegant and useful method in his original derivation. Therefore, it is of interest to fill this gap. Concerning "new physical interpretation" the referee has to note that important work on clarifying and explaining the somewhat dense text of Glauert has been published by J.N. Soerensen and G. van Kuik.

b) Paragraph 2, Lines 13 to 24
It may be of interest to note, that the asymptotic structure $(\lambda \to \infty)$ of $c_P(\lambda)$ has been investigated already in 1955 to forth order in inverse TSR with apparently no connection to Glauert's approach [2].

In addition, it does not seem to be clear what the author means by "very high tip-speed ratios" (TSR). As far as the referee knows, there are a lot of designs with TSR in the order of 10, which may be regarded as a high TSRs.

c) Paragraph 3, Lines 26 to 38
The referee agrees that this regime $(\lambda \to 0)$ is more delicate and - as implied in my statement above - of minor if no practical interest.

d)  Paragraph 4, Lines 40 to 46
Here we also do not agree with the author for reasons also implied in what was written above.

e)  Paragraph 5, Lines 48 to 58
The referee agrees that details of the calculation could have been part of "supplementary material". It would also be interesting to know whether some specific codes (MATHEMATICA, etc.) have been used. Publishing a possible source code would increase transparency. Again, we do not agree that Glauert's model "has no place in any realistic wind turbine analysis".

f)  Paragraph 6, Lines 60 to 68
The paper of Tyagi and S. Schmitz has been reviewed by two experienced scientists. A press release of Sliman (2025) is not under consideration here.

g)  Paragraph 7, Lines 70 – 81
This is a serious point but has already been addressed by the editorial board of Wind Energy Science.

**3.)  Technical Corrections**

This comment is clearly written and needs no technical corrections.

**4.)  References**

[1] J.N. Soerensen, General Momentum Theory for Horizontal Axis Wind Turbines, Springer (2016)

[2] G. Schmitz, Theorie und Entwurf von Windrädern optimaler Leistung (Theory and Design of Wind Wheels with optimum Power), pp 379 – 391, Wiss. Z. d. Univ. Rostock, Germany (1955)

[3] J.G. Leishman, Principles of Helicopter Aerodynamics, 2nd Ed., Ch. 13, Cambridge University Press (2006)

---

## Referee Comment (RC3)

Prof. Gordon Leishman's comment reviews and critiques the paper *"Glauert's Optimum Rotor Disk Revisited – A Calculus of Variations Solution and Exact Integrals for Thrust and Bending Moment Coefficients"* by Tyagi & Schmitz, which was accepted and published in the *WES* journal in 2025. In his comment, Prof. Leishman raises concerns about the paper's limited new insight, novelty and practical value, as reflected (at least) in the following statements:

*"While (the paper) it ultimately offers a seemingly mathematically rigorous reformulation, it contributes limited new insight to the wind energy field".*

and:

*"Although mathematically elaborate, their derivation of "exact" integrals for thrust and bending moment coefficients has limited practical relevance and does not materially advance rotor theory or wind turbine engineering"*

and:

*"Furthermore, the paper fails even to acknowledge the real possibility of tip losses, finite blade count, profile drag, wake expansion, or non-uniform and yawed inflow"*

Having carefully reviewed the manuscript, the two peer reviews, and Prof. Leishman's comment, I find myself in agreement with his preceding comments/statements that the paper's contribution is incremental relative to Glauert's original work on the actuator disk method and that although mathematically accurate it does not have a substantial practical value. The work follows closely the lines of the original actuator disk method and provides some addendum to this work by analytically investigating the limiting behavior of the method and providing analytic expressions for the optimum rotor loads. I would even argue that the use of the term "amendment" is inappropriate, as it implies a revision of the original method rather than an elaboration. Nevertheless, even if the contribution is modest and does not introduce new insights or directions for actuator disk theory—serving instead to expand upon the existing formulation—the crucial question for the review process remains whether the contribution is indeed novel and whether it deserves attention and broader dissemination within the wind energy community. The answer to this question ultimately determines the decision to accept or reject the paper. On this point, considering the large and active research community within the wind energy sector that continues to explore theoretical aspects and variations of actuator disk–based models developed/originated by Glauert, I would not oppose publication. My reasoning is based on the fact that the mathematical derivations presented in the paper represent original work that, to the best of my knowledge, has not been published elsewhere by other researchers. Admittedly, a point of concern is the high overlapping degree of the present work with the authors' prior conference paper; however, I defer to the editorial board's judgment on this matter.

In summary, the paper does not constitute a major advancement of the actuator disk method and does not address limitations inherent in Glauert's original assumptions. However, it provides a mathematically rigorous elaboration on aspects of the original method that have not previously been treated in the literature and, in this respect, is of interest to the wind energy community. Using Prof's Leishman own words, "while the work may be of limited academic interest to those studying the historical development of rotor theory," I would assure

him that, to the best of my knowledge, this interest is by no means limited within the wind energy academic community.

In the second paragraph of the comment, the author of the comment addresses the relationship between the circumferential induction factor and the torque generated by the rotor. This is my primary point of disagreement with his criticism, as I believe the author of the comment misinterprets the way the paper employs the notation $\lambda$. The authors, on multiple occasions, either the notation $\lambda$ or $\lambda\_r$. The former denotes the operational tip speed ratio of the rotor, while the latter represents the local tip speed ratio, referring to the non-dimensional radial position along the blade span rather than the rotor's operational state.

To obtain any integral quantity of the rotor for a given operational condition (as indicated by the $\lambda$ value), one must integrate the radial distribution of $\lambda\_r$ from 0 to the respective $\lambda$ value. Therefore, the tendency of $a'$ towards zero in Figure A1 (the figure implied by the author in his comment) does not imply that $a'$ approaches zero as $\lambda$ tends to infinity; rather, it indicates that $a'$ tends to very small values toward the tip of the blade. As seen in the figure, $a'$ remains significant near the blade root. It should be noted that this refers to the dimensionless axial induction coefficient and not to the swirling velocity toward the tip, which, when multiplied by $\omega \cdot r$, becomes non-negligible. Furthermore, it only refers to the optimal induction distribution.

By integrating the local moment distribution along the blade span, which depends on the local $a'$ values, one can obtain a consistent calculation of the rotor torque. If the distribution of $a'$ is that of figure A1 then torque is maximized. To further support this point, I would add that it can be readily shown that applying the angular momentum equation—which is used to calculate rotor torque— is equivalent to applying Bernoulli's law in the rotating frame (as done in the paper and expressed by Eq. (4)), yielding exactly the same $a'$ distribution as that calculated in the paper when the flow assumed inviscid. I would also add that as earlier mentioned, the theoretical background of the work presented in the paper does not deviate from the original Glauert method. Therefore, obtaining zero torque values for very high $\lambda$ conditions would indicate an unknown limitation of the original actuator disk method.

Based on the above, I would recommend the author to revise his following comment:

*"The authors incorporate swirl into their equations through the angular induction factor; however, in the high-l limit, they assume a'->0, implying that the turbine imparts no swirl and, consequently, no torque. This issue introduces a fundamental contradiction in their theory, i.e., a turbine cannot extract power without torque. While the integrals they have derived may be mathematically consistent, they do not apply to a physically realizable situation."*

The comments in the third and fourth paragraphs are reasonable and are consistent with the discussion presented at the beginning of my review. They address the practical applicability and relevance of the work to the design of modern wind turbines. Once again, the key question is not whether the paper provides groundbreaking information that will directly advance turbine design, but whether it presents material that is of interest and value to the scientific community.

I agree with the comment in the fifth paragraph concerning coefficients and decimals in the derived expressions and would recommend that the journal issue guidelines to ensure the

transparency and reproducibility of the derived equations. I do not though agree with the last comment in the same paragraph:

*"Their model assumes a wind turbine with an infinite number of blades that have a continuously optimal span loading at any λ. This is a mathematical abstraction that has no place in any realistic wind turbine analysis, particularly in the high and low-λ regimes that the authors specifically emphasize."*

The paper addresses the calculation of the loads of rotors (not a single rotor) —specifically, the integral rotor thrust and bending moment—with infinite number of blades, optimized for maximum torque operation at various $\lambda$ values. The expressions for the integral rotor loads derived in the paper refer to the respective optimum/design $\lambda$ value. Naturally, the actuator disk method, when combined with the blade element method, can also be applied to calculate rotor loads at off-design operating points, as the method itself imposes no such limitation. In summary, the information provided in the paper is the loading of different rotor designs (optimized for different $\lambda$ values) at their design point and not the loading of one single rotor that operates optimally at all $\lambda$ values. I would therefore recommend the author to revise his comment.

The comment in the final paragraph has already been addressed by the editorial board, so I have nothing further to add. For the sake of completeness, I would only note that the author of the comment is, of course, correct.

Turning to the comment in the sixth paragraph, I must admit that this is the most delicate point and deserves some discussion and attention. In the text of the paper, the authors never claim to present a groundbreaking work that would open new horizons or possibilities in rotor design. In my humble scientific opinion—and only if I am entitled to judge as I was not a reviewer of the original paper—this is indeed not the case. I hope the authors acknowledge this fact; otherwise, any further discussion on the matter would be pointless. Even the use of the term "amendment" instead of "addendum" appears to be inadvertent, probably without any intention to overestimate the significance of the work (at least it seems so). On this point, I can defend both the authors and the review team. However, I cannot agree with, nor endorse, any decision to promote scientific work in the way this particular study has been publicized in the press and media. Therefore, I agree with the comments of the author in this paragraph.

Whether the journal should implement a specific policy in similar cases in the future is, of course, a matter for the editorial board. That said, the adoption by the journal of the "peer reviewed comment" mechanism is already a step in the right direction.

---

## Referee Comment (RC4)

*Author: "I acknowledge the mathematical distinction between the global tip-speed ratio λ and the local value λ_r. But this distinction the reviewer makes is flawed and departs from the underlying assumption of momentum theory, where λ is treated as uniform across the disk. Introducing λ_r as a spanwise parameter effectively muddles actuator-disk theory with blade-element concepts, which is inconsistent with the theory's own foundation."*

In his original work [1], Glauert distinguishes between **axial momentum theory** (Chapter II of [1]) and **generalized momentum theory**. In the former, the rotor is modeled as a uniformly loaded, perforated disk, with purely axial flow (neglecting swirl). Even within the framework of axial momentum theory, Glauert introduces the concept of the annular element and the spanwise variation of induction. In the generalized momentum theory (Chapter III of [1]), wake rotation is taken into account. The analysis is again carried out using infinitesimal annular elements, but now introduces the concept of a radially varying circumferential induction coefficient. Several recent developments (see, for example, [2], [3]) build on Glauert's generalized theory and extend it to the case of wind turbine rotors — whereas Glauert's original formulation was developed and presented in [1] for aircraft propellers. The paper commented on by the author of the comment is likewise based on the theoretical developments found in these references.

Therefore, I maintain that the distinction made in my comment is valid and not flawed.

[1] Glauert, H.: Airplane Propellers, in: Aerodynamic Theory, edited by Durand, W., Springer, Berlin, Heidelberg, Farnborough, England, 169–360, https://doi.org/10.1007/978-3-642-91487-4_3, 1935.

[2] Tony Burton, Nick Jenkins, Ervin Bossanyi, David Sharpe and Michael Graham, *Wind Energy Handbook*, Third Edition, John Wiley & Sons Ltd. Published, 2021.

[3] Sørensen, J. N.: General Momentum Theory for Horizontal Axis Wind Turbines, Springer International Publishing, Switzerland, https://doi.org/10.1007/978-3-319-22114-4, 2016.

*Author: "This reframing does not resolve the issue. Momentum theory is based on the actuator disk model, which is equivalent to an infinite-bladed, inviscid, tip-loss-free rotor with uniform loading. Within this framework, there cannot be a "family" of rotors; there is only a single idealized construct. The reviewer's notion that somehow there are multiple optimal rotors is therefore a contradiction not afforded by the theory itself."*

Again, this statement seems to contradict the existing literature on the subject. In Section IV of [1], Glauert discusses rotor efficiency and develops families of optimum rotors within the framework of his generalized actuator disk theory. Similarly, Burton et al. [2] and Sørensen [3] extend Glauert's work to wind turbine rotors and derive optimal rotor families.

For example, Burton et al. [2] present the theory of optimal design in Section 3.8 and, in Figure 3.26 (p. 76), illustrate how the maximum Cp of a family of optimal rotors varies as a function

of the design tip-speed ratio (TSR). Comparable studies can also be found in Glauert's original work [1] for aircraft propellers (see, for instance, Figure 18, p. 206) and in Sørensen [3] for wind turbine rotors (see Section 5).

*Author: "At the same time, it is evident from the tone of this review, as well as the two others, that the editors have gone to extraordinary lengths to solicit critiques of my Comment, searching for any conceivable flaw."*

I would like to emphasize that I did not make any effort to find flaws in the author's comment. I believe that my assessment was based solely on scientific evidence, and I fully uphold the presumption of impartiality that characterizes reviewers of scientific publications. I consider the comment to be objective in several of its points of critique (as I have already acknowledged), and my sole aim is to help ensure that it is published free from scientific inaccuracies or potential subjective judgments. Ultimately, I believe that the publication of the comment will be beneficial in every respect — both for the community and for the journal — and this is why my only objective is to ensure that it is scientifically sound and accurate.

I sincerely hope that the references I have cited will assist the author in revising his comment.

---

## Chief Editor Comment (CEC3)

For clarity, in response to Dr. Leishman's statement *"I already conceded ground under pressure from the editor-in-chief as a condition for my Comment to be accepted for review in the first place."*, we report here the correspondence with the Author after the submission of his first manuscript.

Carlo L. Bottasso, Editor in Chief

**Carlo L. Bottasso**

| | |
|---|---|
| **From:** | Carlo L. Bottasso |
| **Sent:** | Wednesday, July 16, 2025 9:24 AM |
| **To:** | Leishman, J. Gordon |
| **Cc:** | Veers, Paul; Jakob Mann; Sandrine AUBRUN; Fleming, Paul; Nicolaos Antonio Cutululis; Gottschall, Julia; Athanasios Kolios |
| **Subject:** | Comment on the Tyagi and Schmitz paper |
| **Attachments:** | TyagiSchmitz_comment_by_Leishman w_markup.pdf |

Dear Prof. Leishman,

The Chief Editors have reviewed and discussed your Comment on the Tyagi and Schmitz paper.

The Editors are very pleased that you have taken the time to write a very thoughtful and penetrating assessment of this work. Your submission makes use of the "peer-reviewed comment to a published paper", an option that -until now- has been underutilized in our journal. We are eager to move forward, though this does take us into somewhat new territory. So, we kindly ask you to be patient as we feel our way forward.

The next step will be to invite peer reviews of your Comment, which we expect will broaden the dialogue significantly. The issues you raise are important and should stimulate a conversation within the community, which is precisely the intent of the open-comment feature of Wind Energy Science.

To launch this relatively new process and encourage responses from reviewers that keep the dialogue productive, the Chief Editors unanimously recommend that your submission be modestly revised before review. There is an opportunity at this early phase to set a tone that avoids sending this potentially constructive dialogue down a path that might be more contentious than it needs to be.

A slight re-wording might help in a few instances. Such comments as *"apparent... rigor"*, *"vague commentary"*, or *"an air of... sophistication"* can cause reviewers to get distracted from the substance of the issues, and tend to make authors defensive rather than accepting of well-deserved criticism. Repeatedly addressing *"these authors"* rather than the content makes the criticism seem personal.

The journal's decision to accept a previously published concept as a new article is indeed fair to question. Our plagiarism detection software did not flag the article due to an extensive restructuring and rewording with respect to the previous conference paper. That said, we will tighten up our processes in response to this situation. In particular, we will publish an editorial comment, and we will improve the wording in our author instructions to help prevent similar issues in the future.

However, we feel that explicitly suggesting the authors are guilty of *"serious ethical concerns"* may not be helpful after pointing out the obvious duplication. We try to be generous in our assumptions of intent, and believe that a less defensive dialogue is more likely if such conclusions are left to the reader. (A copy of your submission is attached with a few highlighted phrases that might be reconsidered.)

The Chief Editors are unanimous in welcoming your contribution and are eager to get the reviews started. We look forward to a constructive dialogue – all thanks to your hard work and generous contribution of time and effort to clarify the public record.

We hope our recommendations for slight re-wording are taken as the purely constructive spirit they are intended.

We look forward to your early response.

Carlo L. Bottasso, Editor-in-Chief

[revised manuscript text omitted]

acknowledge the real possibility of tip losses, finite blade count, profile drag, wake expansion, or non-uniform and yawed inflow. These are factors far more worthy of theoretical attention in wind engineering than the perception by these authors of a "100-year-old math problem" (Ref. 2). Indeed, in Glauert's original theory, the limiting behavior was never considered a theoretical or mathematical problem at all, nor was it subsequently considered a problem within wind energy research, either theoretically or practically. Their so-called "math problem," therefore, is only one of their own invention.

Additional issues arise in the presentation of their results. Several equations (e.g., Eqs. 35, 48, and 50) include numerical constants such as 2.5457 and $-13.3272$ without explanation or derivation. While not strictly erroneous, this practice undermines the rigor and transparency of their work. Providing explanations or citing how these constants were computed would have improved the clarity and independent reproducibility of their results. Without clearly stated methods or symbolic groundwork, the derivations appear procedural rather than intellectually motivated, further weakening the credibility of their results and their relevance. Indeed, despite the apparent technical competence of the derivations, their work remains disconnected from the practical challenges and standards of modern wind turbine research. There is no comparison to empirical data, computational results, or reconciliation with other findings. Their model assumes a wind turbine with an infinite number of blades that have a continuously optimal span loading at any $\lambda$. This is a mathematical abstraction that has no place in any realistic wind turbine analysis, particularly in the high- and low-$\lambda$ regimes that these authors specifically emphasize.

While their mathematical work appears internally consistent within the constraints of an idealized model, the model's assumptions physically break down in precisely those regions where their work attempts to provide insight. These authors do not extend Glauert's theory in a way that adds engineering value or new physical understanding. While the work may be of limited academic interest to those studying the historical development of rotor theory, it certainly falls short of the novelty, applicability, and physical relevance expected of contributions to *Wind Energy Science*. Instead, by clinging to an idealized and largely irrelevant theoretical framework, these authors ultimately misrepresent the spirit of Glauert's original contributions to the theory of rotors and wind turbines. Their work most certainly does not "unlock new possibilities in wind turbine design that Hermann Glauert did not consider" (Ref. 2). It is a sad reflection on the current state of academic publishing that such claims could be legitimized and even tacitly endorsed within the pages of an apparently reputable journal.

Finally, it is important to note that the central derivations, coefficient expressions, and conclusions of this article closely match those previously published by the same authors in a publicly accessible conference paper (Ref. 3). That earlier published work contains essentially the same analytical development, including the polynomial forms, integration techniques, convergence analysis, and variational formulation. This 2025 *Wind Energy Science* article does not cite or acknowledge the prior conference version, which raises a major omission that warrants further editorial attention. Of greater concern is that both papers contain substantial verbatim and near-verbatim reuse of text, structure, and phrasing. The core mathematical derivations are essentially the same, and the conclusions are restated with only minor editorial variation. This unacknowledged repetition constitutes self-plagiarism under standard publication ethics, as it recycles previously published material without appropriate citation or disclosure. While this journal article appears more refined on the surface, the failure to cite a substantially overlapping publication, particularly one with a DOI and broad public accessibility, raises serious ethical concerns about proper attribution and transparency in the communication of research findings.

---

## Author Comment (AC3)

**Review #3 Rebuttal**

It is refreshing to see a review from someone who has clearly studied both the article and my Comment in detail. Yet again, there is an admission by a reviewer that the original article is mathematically incremental, modest in its contribution, and of limited practical value. These reviews all reinforce my central point: the results have **no design relevance** because both limits treated are non-operational for wind turbines. Publishing exact integrals in limits where either the physics is invalid or the operating point is non-existent does not advance rotor engineering.

At the same time, it is evident from the tone of this review, as well as the two others, that the editors have gone to extraordinary lengths to solicit critiques of my Comment, searching for any conceivable flaw. One can only wish the original article had been held to the same standard. Had it been, the contradictions, mathematical mistakes, unexplained constants, and lack of physical relevance would have been obvious at the time of submission, and publication would almost certainly not have followed.

**Reviewer:**

The work [the originally published article] is original enough in derivations, modest, and arguably publishable for community interest. Overlap with the prior AIAA conference paper is concerning but left to the board's judgment.

**Rebuttal:**

The reviewer concedes the article is "incremental" and "modest." That alone supports my central argument that the work introduces no new physical insight and has no engineering value. A thorough review of the original article would have brought out the points that have now been disclosed, and almost certainly would have resulted in a decline to publish decision by any reputable journal. "Community interest" cannot substitute for novelty that advances rotor or wind turbine theory or informs design. As for overlap, the problem is not that a journal article follows a conference paper, which is common practice. The problem is the **failure to cite the 2024 AIAA conference paper**, which created the false appearance of novel new work and exclusive originality to the journal editors and its reviewers.

**Reviewer:**

You misinterpreted $\lambda$ versus $\lambda_r$. Figure A1 shows $a'$ tending to small values only near the tip, not globally as $\lambda \to \infty$. Integrated torque remains finite because $a'$ is significant inboard.

**Rebuttal (revised):**

I acknowledge the mathematical distinction between the global tip-speed ratio $\lambda = \Omega R/V_\infty$ and the local value $\lambda_r = \Omega r/V_\infty$. But this distinction the reviewer makes is flawed and departs from the underlying assumption of momentum theory, where $\lambda$ is treated as uniform across the disk. Introducing $\lambda_r$ as a spanwise parameter effectively muddles actuator-disk theory with blade-element concepts, which is inconsistent with the theory's own foundation. This is not a misinterpretation on my part. If momentum theory is applied as originally formulated by Glauert, then in the limit $\lambda \to \infty$ the swirl factor $a'$ tends to vanish, torque tends to vanish, and yet the mathematical formulation still suggests finite power, which is a contradiction exactly as I noted in my original Comment. However, the overarching point is that **real wind turbines never approach this regime and therefore never operate in it**. One can always arrange the algebra to be self-consistent, but this does not confer physical reality. A turbine cannot generate power without torque, and the global asymptotic limit remains a non-physical construct with no engineering meaning.

**Reviewer:**

Your critique of the low-$\lambda$ limit becomes a general criticism of momentum theory. The article merely offers analytic expressions, which may be of interest.

**Rebuttal:**

My point is certainly not a general dismissal of momentum theory but a critique of its **misuse outside its valid domain**. In the low-$\lambda$ regime, the rotor is operating under conditions of stall, turbulence, and powerful three-dimensional effects. The fundamental assumptions of the actuator disk model, i.e., steady, axisymmetric, one-dimensional (uniform), inviscid flow, **collapse entirely under such conditions**.

Presenting "exact" integrals in this regime does not extend the theory; it applies the theory where its assumptions are **invalid**. The fact that one can write down a closed-form solution does not confer physical meaning. No matter how neatly mathematically expressed, their momentum theory results cannot represent the performance of a stalled rotor with recirculating flows. The results, therefore, are disconnected from both experiment and design practice, and therefore have **no engineering value**.

**Reviewer:**

The article does not claim one rotor optimal across all $\lambda$, but rather a family of optimal rotors each tuned to a specific $\lambda$.

**Rebuttal:**

This reframing does not resolve the issue. Momentum theory is based on the actuator disk model, which is equivalent to an **infinite-bladed, inviscid, tip-loss-free rotor with uniform loading**. Within this framework, there cannot be a "family" of rotors; there is only a single idealized construct. The reviewer's notion that somehow there are multiple optimal rotors is therefore a contradiction not afforded by the theory itself. Whether described as one rotor or as many, the construct remains a mathematical abstraction that does not correspond to any realizable wind turbine. The distinction does not restore physical relevance to my original Comment on this matter.

**Reviewer:**

The unexplained constants should have been better handled, and the journal should issue guidelines.

**Rebuttal:**

Agreement noted. But this is not cosmetic. Numerical constants presented without derivation or explanation prevent reproducibility and transparency, so they appear

mysterious, even arbitrary, to the reader. Their publication without explanation is clear evidence of **superficial review**. A genuine derivation by the authors would have explicitly stated: "Evaluating at the lower limit $a = \frac{1}{4}$ gives the constants $-10.5082$ and $-13.3272$." Instead, the authors just give the numbers, and the journal prints them with no provenance. The journal already operates under a maze of confusing "guidelines" and a labyrinth of review procedures, which is unprecedented in my experience. We have already seen how these journal editors do not follow their own guidelines. Ironically, my Comment has been subjected to far more detailed and rigorous scrutiny than the original article itself, which suggests a double standard is another part of the journal guidelines.

**Reviewer:**

The two integration-limit errors are acknowledged.

**Rebuttal:**

Again, this confirms my point. Transforming variables without correcting integration limits is a basic mathematical error. The fact that both mistakes passed into print demonstrates that the review process was cursory.

**Reviewer:**

The media promotion of the article was inappropriate, but the journal cannot control press releases.

**Rebuttal:**

Agreement noted and crucial. A truly independent reviewer would not spend their effort defending the journal's procedures, but the issue remains and requires resolution. Under the banner of *Wind Energy Science* the authors themselves promoted the article as solving a "100-year-old problem" and unlocking "new design possibilities." I counted over 20 outlets that had broadcast this false claim to the general public. This was false information. Glauert never considered his

formulation a problem; it was already accepted that the limits had no practical design relevance. This article was first promoted in a Penn State University press release and then propagated widely across the internet, creating sensationalism around a "problem" that never existed. This misled both the public and the professional community. The journal cannot disclaim responsibility here: by publishing without correction and allowing such coverage to stand unchallenged, *Wind Energy Science* and its editors have given credibility to a false narrative.